# Tracing the Si Dangling Bond Nanopathway Evolution ina-SiN_x_:H Resistive Switching Memory by the Transient Current

**DOI:** 10.3390/nano13010085

**Published:** 2022-12-24

**Authors:** Tong Chen, Kangmin Leng, Zhongyuan Ma, Xiaofan Jiang, Kunji Chen, Wei Li, Jun Xu, Ling Xu

**Affiliations:** 1The School of Electronic Science and Engineering, Nanjing University, Nanjing 210093, China; 2Collaborative Innovation Center of Advanced Microstructures, Nanjing University, Nanjing 210093, China; 3Jiangsu Provincial Key Laboratory of Photonic and Electronic Materials Sciences and Technology, Nanjing University, Nanjing 210093, China

**Keywords:** resistive switching memory, transient current, trap state

## Abstract

With the big data and artificial intelligence era coming, SiN_x_-based resistive random-access memories (RRAM) with controllable conductive nanopathways have a significant application in neuromorphic computing, which is similar to the tunable weight of biological synapses. However, an effective way to detect the components of conductive tunable nanopathways in a-SiN_x_:H RRAM has been a challenge with the thickness down-scaling to nanoscale during resistive switching. For the first time, we report the evolution of a Si dangling bond nanopathway in a-SiN_x_:H resistive switching memory can be traced by the transient current at different resistance states. The number of Si dangling bonds in the conducting nanopathway for all resistive switching states can be estimated through the transient current based on the tunneling front model. Our discovery of transient current induced by the Si dangling bonds in the a-SiN_x_:H resistive switching device provides a new way to gain insight into the resistive switching mechanism of the a-SiN_x_:H RRAM in nanoscale.

## 1. Introduction

As the key hardware unit of neuromorphic computing chips, resistive random access memory (RRAM) is considered the most promising candidate because of its excellent scalability, fast speed, and good endurance [1,2,3,4,5,6]. Among the next generation of RRAMs, silicon nitride-based-RRAM devices have attracted great interest in recent years because of their low operating current, stable switching behavior, and full compatibility with Si-based CMOS integration technology [7,8,9,10,11,12,13,14]. In particular, a controllable conductive nanopathway could be achieved by tuning the Si dangling bond conducting paths in hydrogenated silicon nitride (a-SiN_x_:H) films with different N/Si ratios, which is similar to the tunable weight of biological synapse [12]. The programming current has been successfully reduced to the lowest record in Si-based RRAM. However, with the down-scaling of a-SiN_x_:H films to the nanometer scale, an effective way to detect trap states related to the dangling bonds in resistive switching is a challenge. Here we first report that the transient current is more favorable for observing the trap states in ultra-thin a-SiN_x_:H RRAM with tunable N/Si ratios. The dynamic evolution of the Si dangling bonds was revealed when the device was switched to different resistance states. We analyze the internal formation mechanism of the transient current. In contrast with other techniques [15], the transient current provides a more effective way to analyze the different distributions of dangling bonds at the programming and erasing state, which is crucial to illuminate the dynamic evolution of the conducting paths in trap-dominated RRAM devices.

## 2. Materials and Methods

To fabricate the RRAM device, a P^+^-Si substrate with a low resistivity of 0.004–0.0075 Ωcm was prepared as the bottom electrode. The oxide on the back surface of the P^+^-Si substrate was removed, and a thin Al layer was evaporated to the back side of the silicon substrate, which can reduce the contact resistance. A 7-nm-thick a-SiN_x_:H film was deposited on the substrate in a plasma-enhanced chemical deposition system at 250 °C using silane and ammonia as the reaction gases. The N/Si ratio x was adjusted by varying the flow ratio of silane and ammonia. Subsequently, 100 nm-thick Al was thermally evaporated on the surface of a-SiN_x_:H as the top electrode using a shadow mask with a diameter of 200 μm to form the final Al/a-SiN_x_:H/P^+^-Si device structure. The atomic concentration ratios of N/Si were determined through XPS measurement at a depth of 5 nm from the film surface. The ESR spectra were measured using the Bruker EMX-10/12 system. The microstructure of the sample was revealed by high-resolution transmission electron microscopy (HRTEM) using a Tecnai G2 F20 electron microscope operating at 200 kV. To detect the transient current related to the traps in all the resistance states, an Agilent B1500A semiconductor analyzer was used to generate the bias signal and record the current intensity. The signals were applied to the top Al electrode with the substrate grounded.

## 3. Results

Figure 1a–e shows the schematic illustration of the transient current measurement of an Al/a-SiN_x_:H/P^+^-Si device at different resistive switching states. Following each resistive switching operation, as shown in Figure 1a-1c, the charging process and transient current measurement were carried out, as displayed in Figure 1d–e. All the electrical measurements were carried out in the original system without position and state changes, so the transient current can be used to trace the resistance state evolution. During the charging process, a constant voltage was imposed on the device for 100 s. Then, the bias voltage was removed, and we measured the current flowing through the device immediately. The oscillogram of the applied voltage for the transient current measurement is shown in Figure 1f. A cross-sectional HRTEM photo of Al/a-SiN_1.17_:H/P^+^-Si device is presented in Figure 1g.

Figure 2a–c shows the time-dependent transient current of the Al/a-SiN_x_:H/p^+^-Si devices at the initial state with tunable N/Si ratios from 1.17 to 0.62. The amplitude of the charging voltage is smaller than that of the forming voltage, which ranges from 0.25 V to 1.75 V, to ensure that the devices remain in their initial states. All the charging currents were measured after the charging time of 100 s. It is found that the transient current intensity is enhanced with the charging voltage increasing when the N/Si ratio is fixed. Under the same applied voltage, the transient current increases as the N/Si ratio decreases. The time-dependent transient current is plotted as a log coordinate, and their slopes are −1. According to the demonstration of Dumin [13], if there are a large number of traps generated in oxides, the discharging current of traps will be the main contributor to the transient current, which obeys 1/t time dependence, and the transient current intensity is proportional to the trap density. Here a slope of −1 in the I–t curves for all devices means the transient current exactly fits the 1/t time dependence. As shown in Figure 2d, the transient current intensity increases as the N/Si ratio decreases from 1.17 to 0.62.

To reveal the role that H played in our devices, we analyzed the FTIR spectra of as-deposited SiN_x_:H films and the one annealed at 600 °C in a vacuum, as seen in Figure 2e,f. The intensity of the Si-H peak from the device with the SiN_x_:H films annealed at 600 °C decreases obviously compared with that of the pristine a-SiN_x_:H films. This result means the number of Si-H bonds is reduced after annealing at 600 °C. Because the Si-H bond energy (318 kJ/mol) is lower than that of Si-N (355 kJ/mol), Si-H bonds are easier to be broken and form the Si dangling bonds. The corresponding transient current intensity of the device with pristine a-SiN_x_:H films and the one annealed at 600 °C is displayed in Figure 2g. It is found that the transient current of the device with the SiN_x_:H annealed at 600 °C is much larger than that of the pristine device. The above analysis confirms that Si dangling bonds induced by broken Si-H bonds make the main contribution to the transient current. The relation of the transient current and the Si dangling bond distribution in SiN_x_:H films can be analyzed by the corresponding ESR spectra, which are presented in Figure 2h. A resonance peak with a g value of 2.0042 is detected from the SiN_x_:H films, which is related to the paramagnetic center of the Si dangling bonds. The intensity of the resonance peak increases as the N/Si ratio decreases from 1.17 to 0.62, which is in agreement with the changing trend of the transient current intensity. As reported by Robertson and Mo et al., as-deposited silicon nitride films contain many silicon dangling bonds, and they are amphoteric deep trap centers located near the middle of the band gap [16,17]. Here, after the charging voltage was removed, electrons released by Si dangling bonds formed the transient current. The number of Si dangling bonds increases as the N/Si ratio decreases [12], leading to the enhancement of the transient current under the same charging voltage. In contrast to the other Si/N ratio, the device with x = 1.17 has the smallest number of Si dangling bonds. The transient current can be detected from the device with the smallest number of Si dangling bonds, which ensures that the same changing trend of transient current can also be detected from the other ones with the larger number of Si dangling bonds. Because the transient current intensity reflects the number of Si dangling bonds in an a-SiN_1.17_:H device, we applied it to trace the trap state evolution during the resistive switching process.

The DC sweep of the resistive switching process for an a-SiN_1.17_:H device by continuously applying different reset voltages is shown in Figure 3a. The device switches from the initial resistance state (IRS) to the low resistance state (LRS) when the magnitude of forming voltage is increased to 4.1 V, with a compliance current of 10 µA. It is worth noting that the current of the Al/a-SiN_x_:H/P^+^-Si device decreases gradually with the reset voltage increasing, which is similar to the tunable weight of a biological synapse. Before the forming process, the transient current was measured after a charging voltage of 0.5 V for 100 s. The low voltage of 0.5 V ensures that the Si dangling bonds can be charged without damage to the Si-H bonds. Following the forming operation, the transient current of the LRS was measured, as shown in Figure 3b. The transient current obviously increases from 1.1 to 10.5 pA after discharging for 0.3 s, which indicates that a large number of Si dangling bonds are generated after the forming process. Subsequently, the reset1 operation made the device switch from the LRS to the middle resistance state (MRS1) under a voltage of −1.36 V. The transient current of the MRS1 is slightly reduced compared with that of the LRS. It means that a part of the generated traps was annihilated during the reset1 operation. After the reset1 operation, we carried out the reset2, reset3, and reset4 operations to switch the device from MRS1 to MRS2, MRS3, and the high resistance state (HRS), respectively. In this case, the transient current decreases gradually with the number of reset operations increasing. It reveals that the generated traps were annihilated gradually during the reset operation. Figure 3c shows the endurance characteristics of the Al/a-SiN_x_:H/P^+^-Si device. It can be seen that a stable memory window and good reliability can be maintained after 200 consecutive cycles. As displayed in Figure 3d, the current of HRS and LRS is still equal to that of the initial state after a retention time of 10^5^ s, which shows good retention characteristics. No noticeable degradation is observed in either state. Figure 3e presents the statistical distributions of SET/RESET currents of 26 devices. It is obvious that the operating voltages of these devices show good uniformity with small deviations.

The discharging of Si dangling bond centers can be described by the tunneling front model, in which at a given time *t*, the tunneling rate is sharply peaked spatially at a depth *x*(*t*) from the Al/a-SiN_x_:H interface [13,18,19,20,21,22]. The traps located closer than *x*(*t*) are emptied, with those beyond *x*(*t*) still occupied. The tunneling front depth *x*(*t*) increases logarithmically with time, as
(1)$$x\left(t\right)=\frac{1}{2\beta }ln\frac{t}{t_{0}},\;\;\;\;\;\;\;\;\;\;\;\;\;\;\;\;\;\;\;\;\;\;\;\;\;\;\;\;\;\;\;\;\;\;\;\;\;\;\;\;\;\;\;\;\;\;\;\;\;\;$$
where $t_{0}$ is the characteristic tunneling time and β is a tunneling parameter that can be calculated as
(2)$$\beta =\sqrt{\frac{2m_{t}^{*}}{\hbar^{2}}E_{t}},$$
where $m_{t}^{*}$ denotes the whole effective mass and $E_{t}$ denotes the trap energy level with respect to the top of the a-SiN_x_:H valence band. Then the transient current can be determined by calculating the velocity of the tunneling front as
(3)$$I\left(t\right)=qnvA=qN\left(x\left(t\right)\right)\frac{dx\left(t\right)}{dt}A=\frac{qN\left(x\left(t\right)\right)A}{2\beta t},$$
where *q* is the electronic charge, *n* is the carrier density, *v* is the tunneling front velocity, *N*(*x*(*t*)) is the spatial distribution of traps, and A is the area of conductive filamentary paths. In this way, we can estimate the lower limit of dangling bond density. In this study, $t_{0}$ is estimated to be 10^−13^ s [18,22], $m_{t}^{*}$= 0.42*m*_0_ [23], and $E_{t}$= 3.1 eV [17]. According to this model, the trap distribution in the a-SiN_x_:H films can be obtained from Equation (3) as
(4)$$N\left(x\left(t\right)\right)=\frac{I\left(t\right)2\beta t}{qA},$$

In Figure 2a, the slope of the log *I* versus log *t* plot is −1, so the original Si dangling bond trap distribution is uniform in the IRS. Similarly shown in Figure 3b, the slopes of the transient current of the LRS and HRS are also close to −1, indicating that the generated Si dangling bond distribution is uniform. According to Equation (4) and the data in Figure 3b, the discharged original Si dangling bond density in the a-SiN_1.17_:H device is calculated to be larger than 10^17^ cm^−3^, while the Si dangling bond trap density generated in the LRS is calculated to be larger than 10^18^ cm^−3^. Here the change in the transient current of the LRS and HRS is related to the generation and re-passivation of Si dangling bonds during the RS process [24,25]. The conducting filament evolution between the IRS, LRS, and HRS is illustrated in Figure 4a–c. As for the forming operation, the conducting paths are composed of Si dangling bonds because the weak Si-H bond can be broken by the forming voltage. The abundant newly generated Si dangling bonds result in a significant increase in the transient current following the set operation. After the reset process, the Si dangling bond conducting paths are partially broken because some Si dangling bonds are passivated by H. Thus, the transient current intensity decreases slightly. When the device switches back to the LRS, some Si dangling bonds are produced due to the broken Si-H bond. Thus, the transient current intensity increases slightly. As displayed in Figure 4d,e, electrons were trapped in Si dangling bonds during the charging process. After the bias voltage was removed, the electrons were released from the trap centers, which is the origin of the transient current.

Figure 5 shows the energy band diagrams of the charging and discharging process, respectively, for an a-SiN_1.17_:H device in different resistance states. In the IRS, thermally-excited electrons are injected into the a-SiN_1.17_:H film from the P^+^ electrode and trapped in these original Si dangling bond trap centers during the charging process. After the removal of the bias, the electrons tunnel from the trap centers to the P^+^ electrode to form the transient current. For the LRS, additional electrons can be captured by the newly generated Si dangling bonds. When the bias voltage is removed, the transient current from both the original and generated Si dangling bonds forms the transient current. As a result, we observe a clear increase in transient current after the forming process. In the HRS, a relatively small number of electrons are captured by the residual Si dangling bond traps due to the passivation of Si dangling bonds by H, and the transient current can be reduced accordingly.

## 4. Conclusions

In summary, we successfully observed the evolution of Si dangling bond nanopathway in a-SiN_x_:H RRAM devices at multiple resistance states by the transient current. The relationship of the transient current with Si dangling bonds for all resistive switching states is revealed in detail. Moreover, the density of the original and generated Si dangling bonds after the forming process can be qualitatively described based on the tunneling front model. The transient current is promising to trace the dynamic evolution of the conducting filament for trap-dominated resistive switching memory, opening a new avenue to insight into the resistive switching mechanism of the a-SiN_x_:H RRAM in nanoscale for neuromorphic computing in the AI period.

## Figures and Tables

**Figure 1 nanomaterials-13-00085-f001:**
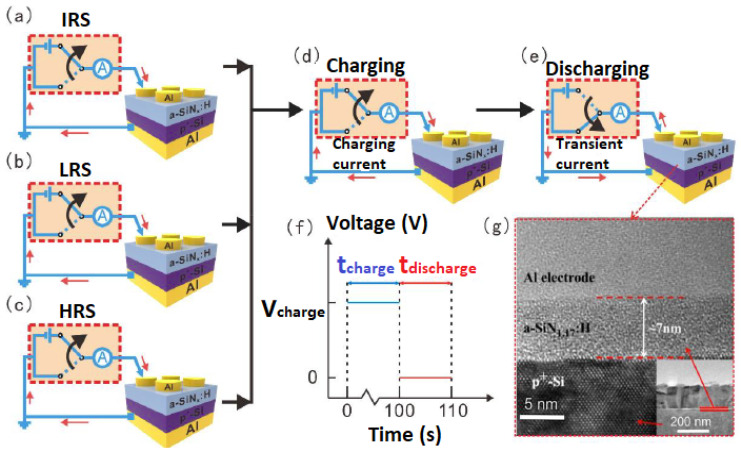
(**a**–**e**) Schematic illustration of the transient current measurement of an Al/a-SiN_x_:H/P+-Si at different resistive switching states. (**f**) Oscillogram of the applied pulse for the transient current measurement. (**g**) Cross-sectional HRTEM photo of the Al/a-SiN_1.17_:H/p+-Si. The inset shows the low-resolution TEM photo.

**Figure 2 nanomaterials-13-00085-f002:**
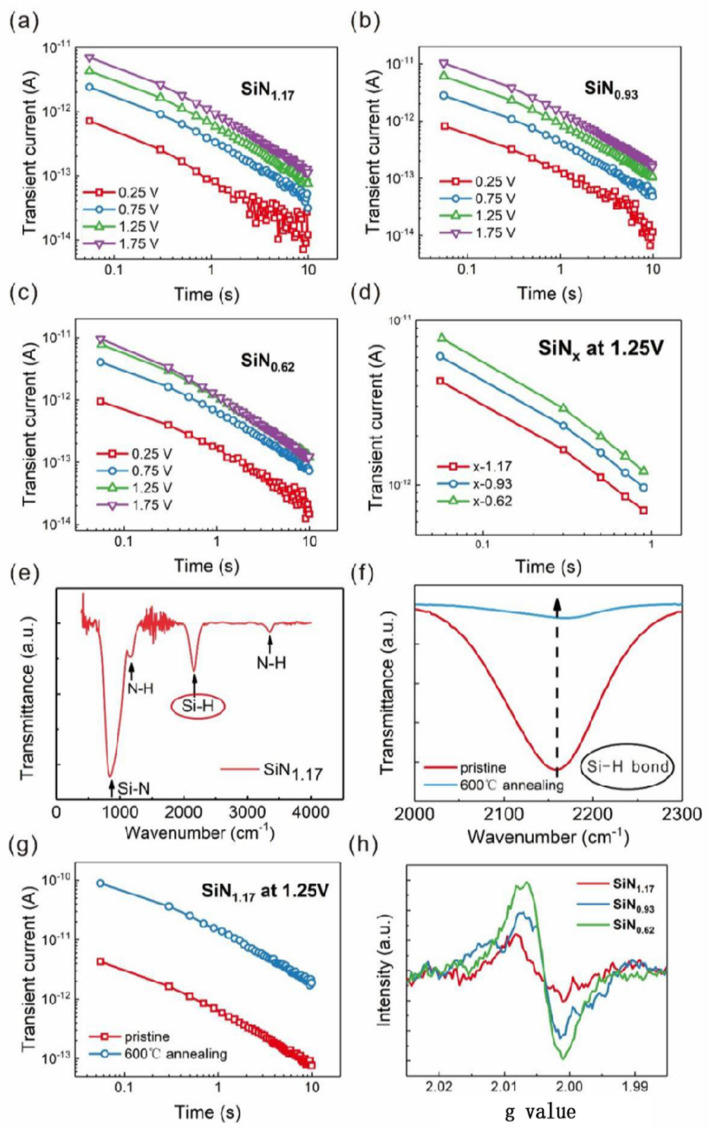
(**a–c**) Time-dependent transient current of the devices at the initial state with N/Si ratios of 1.17, 0.93, and 0.62. (**d**) Time-dependent transient current of the devices with N/Si ratios of 1.17, 0.93, and 0.62 at 1.25V. (**e**) The FTIR spectra of pristine a-SiN_1.17_:H films. (**f**) The evolution of Si–H absorption peak of a-SiN_1.17_:H in the magnified scale before and after 600 °C annealing. (**g**) Time-dependent transient current of a-SiN_1.17_:H before and after 600 °C annealing. (**h**) ESR spectra of the devices at the initial state with N/Si ratios of 1.17, 0.93, and 0.62.

**Figure 3 nanomaterials-13-00085-f003:**
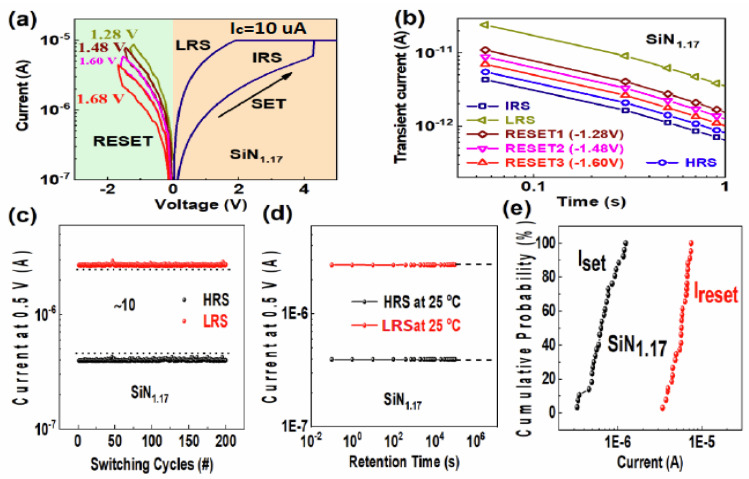
(**a–b**) Multilevel resistive switching I-V curves and the corresponding transient current of a-SiN_1.17_:H-based RRAM device in different resistance states. (**c**) The endurance characteristics of the Al/a-SiN_x_:H/P^+^-Si device after 200 consecutive cycles. (**d**) The retention properties of the Al/a-SiN_x_:H/P^+^-Si device at room temperature. (**e**) The statistical distributions of SET/RESET currents of 26 devices.

**Figure 4 nanomaterials-13-00085-f004:**
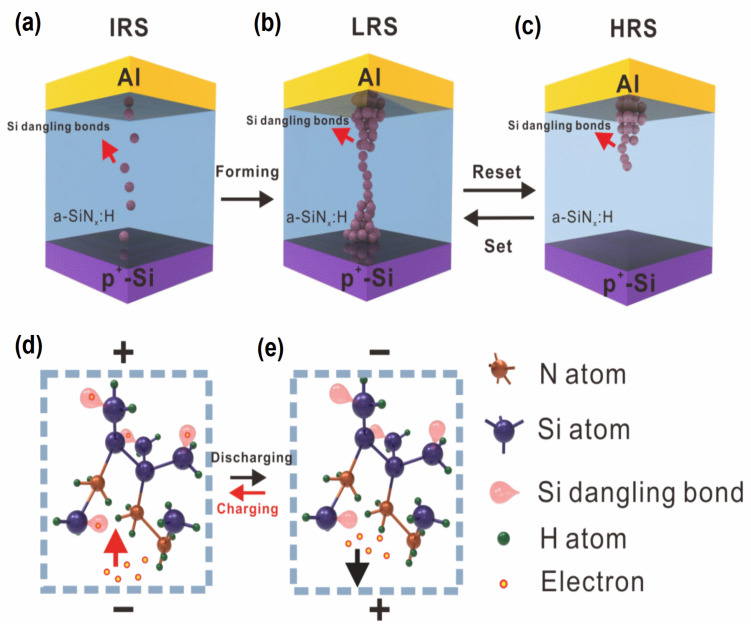
(**a**–**c**) The Si dangling bond distribution of the a-SiN_1.17_:H device in the IRS, LRS, and HRS. (**d**–**e**) Schematic illustration of electrons trapped and released by Si dangling bond during the charging and the discharging process.

**Figure 5 nanomaterials-13-00085-f005:**
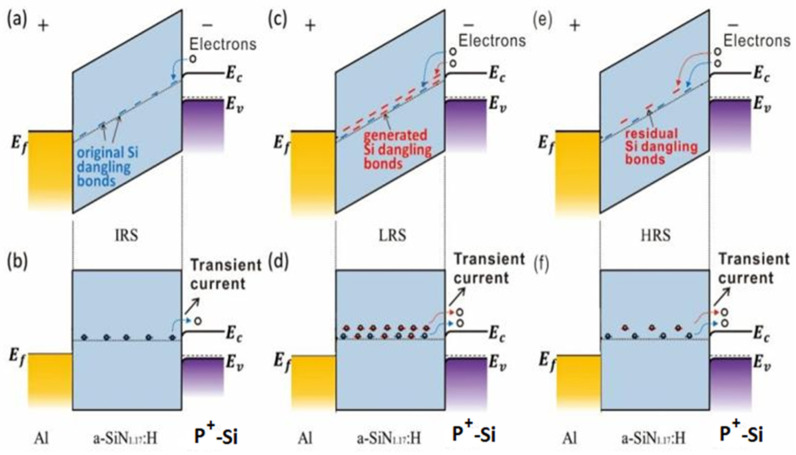
(**a**–**f**) The energy band diagrams of the Al/a-SiN_1.17_:H/P^+^-Si device in the IRS, LRS, and HRS during the charging and discharging process.

## Data Availability

The data that support the findings of this study are available from the corresponding authors upon reasonable request.

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
