# Peer review of "Tracing the Si Dangling Bond Nanopathway Evolution ina-SiNx:H Resistive Switching Memory by the Transient Current"

_nanomaterials, 2022, doi:10.3390/nano13010085_

Round 1

Reviewer 1 Report

Characterizing high and low resistance states is an interesting idea. The paper can be published after minor corrections.

“similar to the tunable weight of biological synapse ” In the case of synapse,  the values of weights are adaptable to the stimulating history. The considered structure has 2 levels while weights of synapse vary in a continuum.

In figure 2h, the horizontal axis unit is Gauss but g has no unit. Gauss is the unit of B. The frequency is F = (1/2π) (e/(2m)) g B. Usually in an ESR experiment, F is given and B is measured. So g varies as the inverse resonant field. 

(318 kJ/mol) is much lower than that of Si-N (355 kJ/mol)” , 318/355 is not much lower than 1

Reviewer 2 Report

The authors present a-SiNx:H memristors providing three memory states controlled by the transient current. Its mechanism is comprehensively investigated. The manuscript is well written, and the contained results are interests to the field of memristors. I would recommend its acceptance for publication in Nanomaterials after properly addressing following comments:

(1) Endurance test including cycle-to-cyle variation (at least 50-100 times) should be given. 

(2) In Figure 1, it seems that the bottom Al electrode was used. But, the authors mentioned that p-Si was used for the bottom electrode. Please clarify it. 

(3) Uniformity test including device-to-device variation should be given. 

(4) Retention test should be included to provide its filament formation in the proposed memristors. 

Round 2

Reviewer 2 Report

The authors have addressed all the comments raised by this reviewer. It seems that the manuscript can be accepected for publication.